# Development and Validation of a New Measure of Work Annoyance Using a Psychometric Network Approach

**DOI:** 10.3390/ijerph19159376

**Published:** 2022-07-30

**Authors:** Nicola Magnavita, Carlo Chiorri

**Affiliations:** 1Postgraduate School of Occupational Health, Università Cattolica del Sacro Cuore, 00168 Rome, Italy; nicola.magnavita@unicatt.it; 2Department of Woman, Child & Public Health Sciences, Fondazione Policlinico Universitario A. Gemelli IRCCS, 00168 Rome, Italy; 3Department of Educational Sciences, University of Genova, 16128 Genova, Italy

**Keywords:** work annoyance, work ability, work engagement, job attitude, work strain

## Abstract

Existing measures of the impact of job characteristics on workers’ well-being do not directly assess the extent to which such characteristics (e.g., opportunity to learn new skills) are perceived as positive or negative. We developed a measure, the Work Annoyance Scale (WAS), of the level of annoyance that workers feel about certain aspects of the job and evaluated its psychometric properties. Using archival data from two cohorts (*n* = 2226 and 655) of workers that had undergone an annual medical examination for occupational hazard, we show the usefulness of the network psychometric approach to scale validation and its similarities and differences from a traditional factor analytic approach. The results revealed a two-dimensional structure (working conditions and cognitive demands) that was replicable across cohorts and bootstrapped samples. The two dimensions had adequate structural consistency and discriminant validity with respect to other questionnaires commonly used in organizational assessment, and showed a consistent pattern of association with relevant background variables. Despite the need for more extensive tests of its content and construct validity in light of the organizational changes due to the COVID-19 pandemic and of an evaluation of the generalizability of the results to cultural contexts different from the Italian one, the WAS appears as a psychometrically sound tool for assessment and research in organizational contexts.

## 1. Introduction

This work is grounded in a substantive-methodological synergy paradigm, as it uses recent methodological developments in a substantively relevant field of research in order to provide a more accurate response to a relevant research question. Specifically, it aims to (a) report on the development and validation of a new measure of work annoyance and (b) demonstrate the usefulness of a network approach [1,2,3] in testing the psychometric properties (i.e., structural and construct validity) of a psychological tool. We start this article by reviewing the key substantive issues that are related to the need for a new measure of work annoyance and then present the methods that we used to provide evidence for its validity.

### 1.1. The Substantive Issue: The Need for a Measure of Work Annoyance

Job attitudes are “evaluations of one’s job that express one’s feelings toward, beliefs about, and attachment to one’s job” (p. 343) [4], and are considered one of the oldest, most popular, and most influential areas of research in all of organizational psychology, given their predictive power with respect to job performance, job satisfaction, and mental health in the workplace [4]. Research has shown that subjective perceptions of job characteristics are situational antecedents of job attitudes and a number of models have been proposed to account for their role in psychological outcomes in the workplace. Hackman and Oldham’ Job Characteristics Theory (JCT) [5] identifies five core subjective job dimensions (skill variety, task identity, task significance, autonomy, and feedback) that predict critical psychological states which, in their turn, predict personal and work outcomes. For instance, Judge et al. [6] found that both objective and subjective indicators of job characteristics were partial mediators of the relationship between core self-evaluations in childhood and early adulthood and later job satisfaction for individuals between the ages of 41 and 50. 

There are two other models that take into account job characteristics to explain health outcomes at work that are the Karasek’s [7] demand/control/support (DCS) model and the Siegrists’ [8] effort/reward imbalance (ERI) model. The former posits that the main sources of job stress come from two basic characteristics of the job itself: job demands and job control (or decision latitude). According to this model, job strain is a function of the interplay of job demand (e.g., quantitative workload, degree of difficulty, time available to perform the tasks, role conflicts, etc.) and the amount of control the worker has over the work (e.g., the ability to make decisions about how to complete job tasks). Social support at work, i.e., the need to relate to others and to seek out help in accomplishing difficult tasks, was later included in the model once its moderating effect on job strain became clear [9,10]. The ERI model takes into account the reward rather than the control structure of work, hypothesizing that psychological issues such as distress are due to a high degree of effort that is not adequately rewarded (whence the so-called imbalance) in the form of pay, recognition, status, or career opportunities. A third risk factor in this model is over-commitment, i.e., a motivational pattern of excessive work-related commitment and a high need for approval [11]. While the DCS mainly focuses on the physical aspects of occupational stress, the ERI is more sensitive to stress arising from work relationships and organizational factors [12], but they both seek out the negative consequences of work. A more recent model, the Job Demands–Resources (JD-R) model [13,14] adopts a more positive view of work and focuses on factors that are related to employees’ well-being with the aim of helping to develop a healthier organizational environment rather than reducing the impact of psychological distress factors. The JD-R conceives stress or work tension as the result of the organizational and job demands (e.g., emotional demands, work overload, interpersonal conflict, etc.) that the workers have to cope with and the resources (e.g., supervisor’s and colleagues’ support, opportunities for skill utilization, autonomy, etc.) that are available to workers to face their job requirements and work challenges. As a result, the JD-R posits that working characteristics can elicit two different psychological processes: (1) the demanding aspects of work can lead to constant psychological overtaxing and eventually to exhaustion; and (2) a lack of job resources prevents the achievement of objectives, hence the experiences of failure and frustration [15]. The JD-R model thus encompasses a wide range of factors that can affect, either negatively (demands) or positively (resources), a worker’s well-being. Although the JD-R model has received empirical support [16], there are no specific tools that allow the assessment of its dimensions, as research has always used scales from other instruments.

The questionnaires that are commonly used to assess the dimensions of the DCS and of the ERI models [17,18] ask participants to rate their level of agreement with descriptions of job characteristics that are known to be associated with distress or other undesirable outcomes but do not directly measure the extent to which such characteristics are perceived as positive or negative by the worker. For instance, the item ‘My job requires that I learn new things’ in the Karasek et al.’s Job Content Questionnaire [18] is considered an operationalization of skill discretion, i.e., the opportunity to use skills that are already possessed and to learn new ones. This is commonly considered a positive job characteristic which, differently from decision authority, is negatively associated with burnout [19]. Workplace learning opportunities and the use of cognitive abilities such as problem solving have been found to be associated with higher job satisfaction and well-being [20,21], but, as argued by Viotti and Converso [22], they are activities that require the use of resources and energies that might be considered by some workers more of a nuisance than a resource, differently from job autonomy and social support. This could also explain why some studies found a positive, and not a negative, association between skill discretion and job demands [23,24,25]. Specifically, the effect of skill discretion is beneficial when the job demands are low, while it can become detrimental when the job demands are high, since it requires the use of energies that may be no longer available [22]. This leads to the substantive issue of this work, i.e., the need for a new measure for the factors that determine what we call work annoyance in workers, that is, the subjective evaluation of work characteristics as something causing vexation or nuisance. On the basis of thousands of assessments that were carried out by the first author in his professional activity as a certified occupational physician, we could identify a number of aspects of the job that are commonly complained about and perceived as annoying by workers and used them to develop the Work Annoyance Scale (WAS). Some of them relate to working conditions, e.g., having to work beyond the set hours, night work, commuting to work, physically demanding job, or a stressful environment. Others are related to the cognitive demands of the job, such as having to learn the use of a new electronic device, having to learn a foreign language, having to work hard to solve a job problem, or having to learn new working techniques. All of these factors have been found to be associated with a reduction in workers’ well-being (satisfaction, happiness, work engagement) and/or with an increase in mental health problems (distress, anxiety, depression) [26,27]. It should be noted that the WAS was developed and tested for validity and reliability before the COVID-19 pandemic.

A more thorough description of the WAS can be found in the Materials section below, and the Italian version that was used in this study is reported in Appendix A. Appendix A also shows an English translation that has been provided with the sole purpose of enabling readers who do not know Italian to understand the content of the items. Although the translation of the questions was carefully crafted with the help of artificial intelligence services and reviewed by the authors, before using the English version of the scale, it is necessary to verify the comprehensibility of the questions for a native English-speaking audience and test its psychometric properties.

### 1.2. The Methodological Issue: The Usefulness of Psychometric Network Approach

#### 1.2.1. The Latent Variable Approach

The classical approach to the measurement of psychological variables is to assume that the characteristic that one wants to measure is an unobservable latent variable (construct) that causes the observable behaviors (i.e., answers to test items) and accounts for most of their variance and covariance—net of the effect of behavior-specific factors. Item scores are thus seen as reflecting (whence the term ‘reflective measurement models’) the position of the individual on the construct—a person who is very extraverted or intelligent will tend to obtain high scores on the items of extraversion or intelligence tests. This assumption raises the question whether psychological constructs actually exist prior and independently of the test that is used to measure them, whether they have a quantitative structure, and whether variations in them actually produce variations in the outcomes of the measurement procedure [28]. The debate is more than a century old and reporting on it is beyond the scope of this work, but there are several works that tackle this issue from different perspectives [29,30,31]. From a methodological point of view, the measurement process that is defined as ‘the assignment of numerals to objects or events according to rule’ (p. 677) [32] applies firstly to the observable behaviors that are consistent with the theoretical definition of the construct, i.e., represent the so-called ‘operationalizations’ (operations that anyone can perform [33]) of the construct. A psychometric model (e.g., a factor analytic or an item response model) will then be used to derive the score on the construct from these observed scores. For instance, if extraversion is defined as ‘an orientation of one’s interests and energies toward the outer world of people and things rather than the inner world of subjective experience’ [34] and extraverts are relatively outgoing, gregarious, sociable, and openly expressive people, an extraversion questionnaire will ask individuals to rate themselves about how much, e.g., they like parties, are talkative, and are full of energy. Next, these ratings will be statistically modeled to obtain a measure of extraversion. As pointed out by Christensen et al. [3], this approach implies that the variation in how individuals answer test items depends on the quantity of the construct they possess and on differences in this quantity—despite the processes that lead different individuals to provide the same answer to the same item might differ [35]. From this perspective, constructs are categories for ‘classifying functionally equivalent forms of behavior in a general population of people’ (p. 347) [36] and are between-person or population attributes that are not necessarily possessed by anyone in the population but can present between-person differences in the population [37]. As a result, the construct can be seen as a descriptor for comparing individuals instead of a causal explanation of their behaviors [38], and thus as a ‘summary statistic’ of the shared variance between test items [3].

#### 1.2.2. The Network Approach

The network approach builds on complex theory [39], and assumes that psychological characteristics are systems, i.e., they are made up of many components that interact with one another, and that they are complex in the sense that their interactions with other systems are difficult to predict due to their interdependencies and properties. In other words, psychological characteristics are not components of a system in terms of latent common causes of behaviors, but features of the system as a whole, consisting of states or stable organizations of dynamic components that reinforce one another [37,40]. This implies that behaviors that are associated with a characteristic can directly influence (and/or be influenced by) behaviors of another characteristic, as they do not actually measure the characteristic but are part of it. Therefore, psychometric tools measure parts of the characteristic, not the characteristic itself [1,41]. Specifically, they refer to the state of a specific group of components that are causally dependent on one another and form a network whose state is determined by the total activation of these components [42]. For instance, the more components of depression are active, the more the network is driven toward depression as a stable state. Thus, a psychological characteristic is no longer a between-person attribute as in the classical approach, but a within-person network that evolves in time [41]. 

In this perspective, the meaning of the test items and the explanation of their association are different from the classical approaches, such as the one that is implied by factor analysis. Factor analysis assumes that item score intercorrelations can be explained in terms of one or more underlying, distinct, and unique causes. For instance, the Five Factor approach to personality [43] posits that individual differences in adult personality characteristics can be organized in terms of five broad trait domains. Personality questionnaires that were developed in this context comprise of groups of items that (are thought to) reflect these domains, so that each group should be associated with a unique causal system. However, subgroups of items in a domain are often more strongly associated with each other than with the others in the same domain, reflecting narrower characteristics that are commonly referred to as facets. As a result, a domain can comprise of both items and facets [44]. In the network approach, some facets may reflect a unique cause, while others may reflect heterogeneous causes, which must be separated into unique components, i.e., items or sets of items that share a unique common cause. These components correspond to attributes in the classical approach, and the characteristics that are assessed by existing psychometric tools can be conceived as composite attributes [35], i.e., finite universes of attributes comprising of a limited number of unique attributes [45]. While some factor analytic approaches (e.g., the independent cluster model in confirmatory factor analysis [CFA]) assume that attributes of a characteristic exist independently of other characteristic domains, the network approach accommodates for so-called “fuzziness” [40], i.e., an overlap of attributes with other domains.

This has important implications for the validation process of a psychometric tool. Since attributes do not directly measure the characteristic but its parts, the validity of the tool is supported by the evidence that the tool measures the state of the network that is composed of causally connected components. The response processes in network models are assumed to lie in the reciprocal cause and effect processes of other components, and not in the causal effect of the latent variables. This also implies that one should specify how the response process of one component has reciprocal causes and effects on the processes of other components [3].

Recently, Christensen et al. [3] have proposed a procedure for establishing evidence of structural validity of a psychometric tool through item, dimension, and internal structure analyses and suggested how external validity could also be tested, still from a network perspective. Their approach is complementary to classical latent variable models, as it assumes a different data-generating mechanism, and can provide additional information about the relationships between the observed variables. Such an approach comprises of three steps: redundancy analysis, dimension analysis, and internal structure analysis.

##### Redundancy

Redundancy is a somehow overlooked issue in the development of psychometric tools, which appears to be caused by the researcher’s will to meet some desirable statistical property without paying attention to the item content. For instance, if one wants to develop a relatively short scale that is unidimensional and highly internally consistent, chances are that they will end up with a set of highly correlated items that are basically the same item written over and over in slightly different ways [46], because this would ensure an optimal fit in a CFA and a high Cronbach’s alpha. Such high specificity is adequate if the construct to be measured has a narrow conceptual breadth (e.g., angry hostility), but not if the construct has a greater breadth (e.g., neuroticism). Since in the network approach characteristics are composed of unique causal components and are not exchangeable with other components of the system [41], such components have to be unique rather than redundant to reduce latent confounding [47]. Christensen et al. [3] proposed assessing redundancy using weighted topological overlap [48], which is a measure of the similarity of two nodes’ (i.e., items’) connections to other nodes, i.e., it quantifies the similarity between the magnitude and direction of two nodes’ connections to all other nodes in the network. The larger the topological overlap of two items, the greater the shared functional or latent influence. Christensen et al. [3] devised a method to determine which node pairs overlap significantly with one another and implemented it in the UVA function in the EGAnet R package [49]. Although the authors suggest removing all but one item or combining items into a single variable when redundancy is detected, this cannot be a substitute for an accurate check of item content, since sometimes items can actually tap into conceptually different content despite their statistical redundancy.

##### Dimensionality

Dimension analysis addresses the same issue as traditional exploratory factor analysis (EFA) methods from a network perspective. Specifically, it aims at identifying the number of so-called communities in the network by maximizing the connections within a set of nodes while minimizing the connections from the same set of nodes to other sets of nodes in the network through community detection algorithms [50]. Although communities are conceptually similar and statistically equivalent to factors [51], they do not necessarily represent a common cause, but they are dimensions that emerge from densely connected sets of nodes that form coherent subnetworks within the overall network.

Dimension analysis can be performed through exploratory graph analysis (EGA) using the EGA function in the EGAnet R package [49]. EGA initially estimates a Gaussian graphical model (GGM) [52,53], using the graphical least absolute shrinkage and selection operator (GLASSO [54]) or the triangulated maximally filtered graph approach (TMFG [55,56]), in which the edges represent (regularized) partial correlations between nodes after conditioning on all other nodes in the network. Then, it uses the Walktrap community detection algorithm [57], which determines the number and content of communities in the network through random walks [56]. It has been shown that EGA performs equally well or better in identifying the number of dimensions than common factor analytic techniques, such as parallel analysis [56]. Differently from EFA, EGA (1) does not require a rotation method, since the association between dimensions can be assessed through the number of connections between items of one dimension and items of another dimension, thus revealing which items are responsible for the cross-dimension relationships; (2) does not require the researcher to decide on item allocation, as the algorithm puts items directly into dimensions; and (3) allows the researcher to evaluate if some dimensions are more central than others in the network and if the items on the questionnaire are actually organized into the dimensions they expected. An EFA loading equivalent can also be computed. Building on Hallquist et al.’s finding that node strength (i.e., the sum of a node’s connections) is roughly redundant with CFA factor loadings [47], Christensen and Golino [58] developed a measure named network loading, which is the standardization of node strength split between dimensions and more closely resembles EFA loadings. However, while factor loadings are the regression coefficients of the item scores on the score on the factor and represent how well an item is an observable reflection of the factor, network loadings represent each node’s contribution to the emergence of a coherent dimension in the network [58]. Christensen and Golino [58] also suggested that by multiplying the network loading matrix by the observed data, one can obtain a weighted composite for each dimension that is the equivalent of a factor score.

##### Internal Structure

Internal structure analysis assesses whether scales are unidimensional and internally consistent using a measure, structural consistency, that overcomes the issues of common measures of internal consistency, such as Cronbach’s alpha (for a review, see [59]), and is a combination of homogeneity and internal consistency in a multidimensional context. Christensen et al. [3] define structural consistency as the extent to which causally coupled components form a coherent subnetwork within a network, i.e., the extent to which items in a dimension are homogeneous and interrelated given the multidimensional structure of the questionnaire. It can be computed through bootstrap exploratory graph analysis (bootEGA, [60]). The function estimates a GLASSO network from the data and takes the inverse of the network to derive a covariance matrix. Then it uses this covariance matrix to simulate data with the same number of cases as the original data from a multivariate normal distribution and applies EGA to it, determining each item’s assigned dimension. This procedure is repeated until the desired number of bootstrap samples is achieved. This procedure provides a sampling distribution for the total number of dimensions and each item’s dimension allocation. Using the original EGA results, structural consistency and item stability can be computed. The former corresponds to the proportion of times that each empirically-derived dimension is exactly (i.e., identical item composition) recovered from the replicate bootstrap samples, and thus can range between 0 and 1. Item stability is the proportion of times that each item is identified in each empirically derived dimension across the bootstrap samples. This measure allows the researcher to identify which items may be causing structural inconsistency and the other dimension(s) these items are being assigned to. It may emerge that these problematic items replicate only in a new dimension, thus forming a new separate one, or that they replicate more with another dimension, thus fitting better with another dimension, or that they replicate equally across multiple dimensions, thus identifying as multidimensional.

##### External Validity

Christensen et al. [3] also provided suggestions about strategies for testing external validity in a network perspective. Construct and criterion validity are commonly tested using correlational and/or regression methods that test whether and how the measure under examination is related to measures of other constructs in the same nomological net and/or to relevant covariates, such as background characteristics. However, these methods cannot map out multicollinearity and predictive mediation (see., e.g., [61]). Standard multiple regression tests the contribution of each predictor to the prediction of the criterion keeping the other predictors constant. Nevertheless, since one or more predictors are predicted, in their turn, by the other predictors in the model, they will have less and less unique information that can be added to the prediction of criterion. This issue goes by the name of ‘multicollinearity’, and tends to be more and more problematic when the multiple correlation of one or more predictors with the set of other predictors increases, making the estimates of single regression coefficients unreliable and difficult to interpret. This is especially common in cross-sectional studies [62], as is the case here. Some countermeasures exist (see, e.g., [62], Section 10.6), but they do not directly provide insight into predictive mediation, i.e., a network in which two variables are not directly connected but are indirectly connected (e.g., X–Z–Y). This indicates that X and Y may be correlated, but any predictive effect from X to Y (or vice versa) is mediated by Z [61]. GGMs can be a viable solution also to this issue, since the strength of the edges is the partial correlation coefficient that corresponds to a multiple regression coefficient, i.e., an estimate of strength of the direct association of one variable with another after controlling for all other variables in the network—thus ruling out spurious effects due to other variables in the network. In their basic applications (see, e.g., [61]), GGMs allow the investigation of the structure of the network, but it is also possible to compute an equivalent of the R^2^ index in multiple regression when one adds the computation of ‘predictability’, that is, how much variance of a variable is accounted for by the variables that are connected to it [63]. Even newer methods, such as Bayesian networks [64], can also provide insights into the admissible causal relationships in the network building upon the properties of Directed Acyclic Graphs (DAG), which express the conditional independence relationships between variables by using graphical separation. However, rigorous causal inference requires that there is actually a DAG underlying the data, that all causes of a given variable are measured, that all variables that are connected in a given way are probabilistically dependent, and that there are no bidirectional causal relations, assumptions that are hard to meet in psychological studies [64].

### 1.3. The Present Study

In this work we show how the network psychometrics approach can be used to evaluate the redundancy, the dimensionality, internal structure, and external validity of a newly developed psychometric tool. In order to help readers appreciate the similarities and differences of this approach from the traditional one in testing the internal structure and the external validity of the scale, we also show in parallel the results of more common statistical methods, such as parallel analysis, exploratory structural equation modeling (ESEM, [65]), and correlational analyses. In order to provide evidence for the replication of results, we performed the analyses on archival data from two different cohorts of workers that met the recommended sample size (i.e., ≥500) for network analyses [3].

## 2. Materials and Methods

### 2.1. Samples and Procedure

The data for this study come from archival data of the first author. In Italy, workers that are exposed to occupational hazards at their workplace must undergo an annual medical examination by a certified occupational physician, during which anamnestic and physical data are collected. While waiting for their medical examination, workers were invited to fill out a questionnaire including some measures of psychological functioning, which are described below. It should be noted that these measures were part of a screening for occupational issues, such as distress or sleep disturbances, and not aimed at testing research hypotheses. We considered two cohorts (Cohort 1, *n* = 2226; Cohort 2, *n* = 655) of workers from the healthcare sector that were assessed before the COVID-19 pandemic and could provide at least 500 cases for the network analyses, as recommended by Christensen et al. [3]. The descriptive statistics for background, work, and health-related characteristics and questionnaire scores are reported in Table 1.

### 2.2. Materials

#### 2.2.1. Cohort 1

Workers in Cohort 1 completed a survey that included a background schedule and some questionnaires. The background schedule asked them to report on: age; biological sex; whether they worked at night; whether in the last year they had experienced physical assaults, threats, harassment, stalking, and accidents while working; whether they had experienced an accident while at home; whether they had experienced some trauma (e.g., death of a beloved one); whether they had an accident while driving; whether they had come close to an accident while driving; the frequency with which they had dozed off while driving (from 1 = ‘Never or almost never’ to 5 = Almost every day’); the weekly frequency of a 30-min physical activity (form 1 = ‘Never’ to 4 = ‘Three times or more’); the weekly number of alcoholic drinks (i.e., 20 cl/6.76 oz glass of wine or 33 cl/11.16 beer bottle or 40 mL/0.14 oz glass liquor) that were consumed (from 1 = ‘None to 4 = ‘More than 16’); whether they were smokers; the frequency with which they tried to limit salt, sugar, or fat in their meals (from 1 = ‘Never’ to 4 = Every meal’); their level of job satisfaction (from 1 = ‘Extremely unsatisfied’ to 7 = ‘Extremely satisfied’); their level of happiness (from 0 = ‘Very low’ to 10 = ‘Very high’); their current working capacity, compared with the highest working capacity they had had in their life (from 0 = ‘Completely unfit for work’ to 10 = ‘Highest working capacity’). Data for weight, height, hypertension, hypercholesterolemia, hypertriglyceridemia, and hyperglycemia were obtained through appropriate medical examinations. Questionnaires included the following.

Work Annoyance Scale (WAS). The WAS is a nine-item self-reported measure that asks participants to rate the level of annoyance that they feel (or would feel) about certain characteristics of their work activity on an 11-point, Likert-type, intensity scale (from 0 = ‘No annoyance’ to 10 = ‘Utmost annoyance’). The items describe some common working conditions (e.g., having to work at night) and cognitive demands of work (e.g., having to learn a new language) that workers often complain about. The content of the items was derived from anecdotal comments or reports during the annual medical examination by workers belonging to earlier cohorts than those investigated in this study. 

*Effort-Reward Imbalance Questionnaire-Short Form* (ERIQ-SF, [67]). The ERIQ-SF is a 16-item assessment tool for the dimensions of the ERI model: effort (3 items), reward (7 items), and overcommitment (6 items). Participants are asked to rate their agreement with descriptions of job characteristics on a 4-point, Likert-type scale (from 1 = ‘Strongly disagree’ to 4 = ‘Strongly agree’). The items for the Italian version were taken from Magnavita [68]. Higher scores corresponded to higher levels of effort, reward, and overcommitment, respectively.

*Social support scale from the Swedish Demand-Control-Support Questionnaire* (DCSQ-SS; [69]). The DCSQ is a shorter and modified version of Karasek’s [18] Job Content Questionnaire. The Support scale comprises of six items that tap into the climate and quality of interpersonal relations in the workplace. Items are rated on 4-point, Likert-type, agreement scale (from 1 = ‘Not true at all’ to 4 = ‘Completely true’). The items for the Italian version were taken from Magnavita [68]. Higher scores corresponded to higher levels of perceived social support at work.

*General Health Questionnaire-12* (GHQ-12; [70], Italian version in [71]). The GHQ-12 is a list of 12 issues that are related to anxious/depressed states and social dysfunction. Participants are asked to report the frequency with which they have experienced each issue on a 4-point, Likert-type frequency scale (from 1 = ‘Never’ to 4 = ‘More than usual’). Higher scores corresponded to higher levels of distress.

*Pittsburgh Sleep Quality Index* (PSQI, [72]; Italian version in [73]). The PSQI is a 19-item, self-reported measure of sleep quality over a 1-month time interval. It provides scores in seven components (subjective sleep quality, sleep latency, sleep duration, habitual sleep efficiency, sleep disturbances, use of sleeping medication, and daytime dysfunction) that can be added up to obtain a total score. For the purposes of this study, we did not consider the total score. Higher scores corresponded to lower levels of sleep quality.

*Berlin Questionnaire* (BQ, [74]; Italian version in [75]). The BQ is a 10-item screening tool for obstructive sleep apnea syndrome (OSAS) and allows categorization of patients as either high or low risk for OSAS based on self-reports of snoring, daytime sleepiness, hypertension, and obesity. It provides scores in three categories (snoring, daytime somnolence, and hypertension/obesity). In the analyses we did not consider the score in third category as we used objective data for computing the Body Mass Index (BMI) and detecting hypertension. Higher scores corresponded to higher levels of sleep apnea issues.

*Epworth Sleepiness Scale* (ESS, [76]; Italian version in [77]). The ESS is an eight-item, self-report measure for the assessment of sleep propensity in real-life situations. Participants are asked to rate on a four-point, Likert type scale (from 0 = ‘Never’ to 3 = ‘High chance’) their usual chances of dozing off or falling asleep while engaged in routine activities. Higher scores corresponded to higher levels of sleepiness.

#### 2.2.2. Cohort 2

Workers in Cohort 2 were asked to report on their age, biological sex, job satisfaction, happiness, working capacity, desired retirement age (before 60, between 60 and 65, after 60, don’t know), the extent to which they perceived their working environment as an elder-hostile environment (from 1 = ‘not at all’ to 5 = ‘very much’), their perception of their health conditions (from 1 = ‘very bad’ to 5 = ‘very good’). They also completed the WAS, the ERI-SF, the DCSQ-SS, and the Goldberg Anxiety and Depression Scale (GADS, [78], Italian version in [79]). The GADS comprises of two sets of nine items describing common anxiety and depression symptoms, and participants are asked to report whether or not they had experienced each of them in the last few days. Higher scores corresponded to higher levels of anxiety and depression symptoms, respectively.

### 2.3. Data Analysis

Given the nesting of participants in both cohorts into organizations, we considered using a multilevel approach to data analysis. However, a preliminary check of the intraclass correlation coefficients of the WAS item scores showed that items had a negligible between-organization variance, and that the Pearson item correlation matrix that was computed without taking into account the nesting of observations into organizations did not significantly differ from the one that did (i.e., the within correlation matrix, see Appendix A). We performed the analyses that are presented here with both matrices, and the results did not meaningfully differ. Therefore, for the sake of simplicity, we present here only the results that were obtained without taking into account the nesting of observations into organizations.

We initially performed the traditional analyses of dimensionality of the WAS item pool, i.e., scree-test [80], parallel analysis (PA, [81]), and minimum average partial correlation statistic (MAP, [82]). When using the scree-test, the optimal number of factors corresponds to the number of factors before which the downward curve of the eigenvalues seems to flatten out. PA compares the observed eigenvalues to the eigenvalues that are generated from a simulated matrix of random data of the same size. On the basis of the recommendations of Buja and Eyuboglu [83], we performed PA on 1000 random correlation matrices that were obtained through permutation of the raw data, and following Longman et al. [84] we considered the 95th percentile random-generated eigenvalues as the threshold values. 

Velicer [82] suggested that the optimal number of factors is the one at which the average partial correlation of the variables (i.e., the MAP statistic) reaches its minimum after partialling out the factors. These analyses were performed on Pearson correlation matrices using principal component eigenvalues [85,86] through the *fa.parallel* and the vss functions in the R package *psych* [87].

Once the optimal number of factors was determined, we used ESEM to investigate the factor structure for the WAS. ESEM allows for the estimation of all factor loadings (subject to the constraints that are necessary for identification) and, in general, for an exploration of complex factor structures (similarly to EFA) while allowing access to parameter estimates, standard errors, goodness-of-fit statistics, and modeling flexibility (e.g., correlating error variances, obtaining factor scores corrected for measurement error, etc.) usually afforded by CFA [65]. We used the robust maximum likelihood (MLR) estimator to take into account the relative nonnormality of the item score distributions and the goodness-of-fit of ESEM models was evaluated using the robust versions of the comparative fit index (CFI); the Tucker–Lewis Index (TLI); the root-mean-square error of approximation (RMSEA), with its 90% confidence intervals (CI); and the standardized) root-mean-square residual (SRMR). We used the following criteria for model fit [88]: TLI and CFI: values ≥ 0.90 indicated acceptable fit, values ≥ 0.95 indicated excellent fit; RMSEA and SRMR: values ≤ 0.08 indicated acceptable fit, values ≤ 0.06 indicated excellent fit. These analyses were performed using the sem function in the R package *lavaan* (version 0.6-10, [89]).

We then evaluated the redundancy, internal structure, and structural consistency of the WAS items using the functions that were described in Section 1.2. All these analyses were performed on Pearson correlation matrices. Finally, we tested the external validity of the WAS using Pearson correlations and network analysis. In the former, given the large sample sizes, we chose not to consider the statistical significance of the correlation coefficients, since correlations as low as 0.076 (Cohort 1) and 0.140 (Cohort 2) would have resulted statistically significant at alpha = 0.05 and at a power = 0.95 for a two-sided test. We, therefore, chose to rely on the effect size, i.e., following Cohen (1988) we considered as ‘negligible’ correlations < |0.10|, as ‘weak’ correlations in the |0.10|–|0.30| range, as ‘moderate’ correlations in the |0.30|–|0.50| range, and ‘strong’ correlations > |0.50|. For network analyses, we used the nonregularized method of network estimation that was recently developed by Williams and colleagues [90,91,92]. This method addresses some issues that are related to the performance of the GLASSO when the number of cases largely exceeds the number of variables and sparsity (i.e., there are much fewer links than the possible maximum number of links within the network) is not warranted, as it is the case here. Such a method is implemented in the R package *GGMnonreg* (version 1.0.0, [93]), and uses the BIC information criterion and the forward selection method. Bootstrap procedures (10,000 samples) were used to identify edges that could be considered as reliably different from zero, that is, their 95% confidence interval did not contain zero. We also computed the predictability for each node. Before carrying out the analyses, variable scores were transformed as follows, following Gelman’s suggestions [94,95]: metric variables and ordinal variables with more than five categories were shifted to have a mean of 0 and scaled to have a standard deviation of 0.5; each score of ordinal variables with five categories or less was assigned its rank and ranks were transformed into z-scores using the inverse-normal cumulative distribution function and then divided by two, in order to have them on the same metric as the metric variables; binary variables were shifted to have mean of 0 and a difference of 1 between their two categories (i.e., as males were 67%, we defined the centered biological sex variable values to 0.32 and −0.67). In Cohort 1, the body mass index was computed from the height and weight and categorized as underweight (BMI < 18.5), normal weight (18.5 ≤ BMI < 25), overweight (25 ≤ BMI < 30), and obesity (BMI ≥ 30). This variable was then dummy coded with normal weight as reference. In Cohort 2, the desired retirement age was dummy coded with the category “Don’t know” as a reference.

The relevant R code for the analyses used in this work is reported in Appendix A.

## 3. Results

### 3.1. Traditional Approach

The item score distributions and descriptive statistics are reported in Figure 1, and they are very similar for the two cohorts. Figure 2 shows the results of the dimensionality analyses, and all suggested that the optimal number of factors was two. We thus performed ESEM setting the number of factors to be extracted to two, with *geomin* rotation.

The results of the ESEM analysis showed an optimal fit for the two-factor model (Cohort 1: CFI = 0.984, TLI = 0.970, RMSEA = 0.049 [0.040, 0.059], SRMR = 0.017; Cohort 2: CFI = 0.990, TLI = 0.980, RMSEA = 0.040 [0.023, 0.061], and SRMR = 0.018). The factor loadings (as pattern coefficients) and the correlations between the factors are reported in Table 2 and show a clear two-factor structure with moderately correlated factors. However, item 9 has a substantial loading (i.e., >0.30) on both factors. As pointed out by Christensen et al. [3], this is would be a case in which the placement of the item into a dimension would require researchers’ decision. 

### 3.2. Network Approach

We initially performed a redundancy analysis. Traditionally, this analysis could be carried out looking for correlations that were higher than a certain threshold in the correlation matrix. For instance, one might define redundancy between two items as a proportion of 0.50 or more of shared variance, which would correspond to a correlation of 0.71. As shown in Appendix A, the correlation of item 3 with item 5 exceeds this threshold in both cohorts. Another strategy could be testing whether a pair of correlations to a third variable are significantly different from each other, compare every possible combination of correlations in a network, and calculate the proportion of correlations which are significantly different for each different pair of nodes. Using a threshold, the researcher can set the proportion of correlations which is deemed “too low”, and all pairs of variables which fall below this threshold could be considered as redundant (see the *goldbricker* function in the *networktools* package [96] in *R*). A further strategy could be to inspect the residual correlations of a factor model, but this would imply that a model has been fitted to data. The method that was implemented in the *UVA* function in *EGAnet*, instead, identifies possible redundant nodes/variables and the associated “redundancy chains” by computing a clustering coefficient. In our case, we found a redundancy chain between items 3, 5, and 6 in Cohort 1, and between the same items, on the one hand, and items 7 and 8, on the other, in Cohort 2. In these cases, only one item per chain could be retained, or the items could be combined into a single variable. Christensen et al. [3] recommend this latter option in order to retain all the information that is available, but in case of long measures, the former approach looks more parsimonious. The decision about which item to choose could be based on content (e.g., taking the most general case of the attribute) or the item with the higher variance [97] or, we would add, squared multiple correlation. However, the inspection of the item content might make us retain all the items of a chain, since, as in this case, there is no such strong content overlap as to suggest that they should be merged. The items of the three-item chain that was found in both cohorts tap into a general cognitive demand of learning something new, but learning a new technique (item 3), a new electronic device (item 5), or a foreign language (item 6) seem to be different enough to suggest keeping them separate, despite their statistical overlap. 

We then performed dimensionality analysis with the *EGA* function (model = *glasso*, algorithm = *louvain*). The results are shown in Figure 3. As explained above, the function automatically selects the optimal number of factors and assigns variables to dimensions. The results are the same in either cohort, and parallel those of ESEM. In this case, however, item 9 is always placed in dimension 2. Using the function *net.loads* we then investigated each node/variable’s contribution to the coherence of these dimensions by computing its standardized strength for each dimension, i.e., structural coefficients. These values are the equivalent of factor loadings in ESEM, and we reported them in Table 1 alongside the others, but they represent partial correlation, and not zero-order correlation loadings, hence their being systematically smaller than ESEM loadings. Christensen and Golino [58] suggested using 0.15, 0.25, and 0.35 as thresholds for small, moderate, and large network loadings, respectively.

As a measure of both unidimensionality and internal consistency, we computed the structural consistency via the *bootEGA* (10,000 samples) and *dimensionStability* functions. In Cohort 1, the structural consistency coefficients for dimensions 1 and 2 were 1.000 and 0.999, respectively, while in Cohort 2 they were 0.990 and 0.986. These values indicate perfect or almost perfect structural consistency, i.e., the proportion of times that each empirically-derived dimension is perfectly (exact same composition) recovered from the replicate bootstrap samples. Omega reliability coefficients, computed as “omega total” with the *scaleStructure* function from the *ufs* package in R [66], were 0.77 and 0.81, respectively, in Cohort 1 and 0.80 and 0.81, respectively, in Cohort 2 (see Table 1). The average item stability, i.e., the proportion of times that each item is identified in each empirically derived dimension across the replicate samples, was 1.000 for either dimension in Cohort 1, and 1.000 and 0.994 in Cohort 2. The details of stability for each item are shown in Figure 4. As the proportions of replication approach 1, this means that all the items are consistently being identified within their original dimension. Notably, this is true also for item 9, which was the one with a substantial cross-loading (Table 2). Were the proportion of replication of this item substantially smaller, it would have meant that it fell also within the other dimension’s domain.

Based on item content, we named the two dimensions “Working conditions” (Dimension 1) and “Cognitive demands” (Dimension 2). Since practitioners are likely to compute total scores by summing up item scores in each dimension, we assessed the correlation between these observed scores, ESEM factor scores, and network scores that were derived by multiplying the loading matrix by the observed data to derive a weighted composite for each dimension, as suggested by Christensen et al. [3]. The results are reported in Table 3 and indicate almost perfect overlap among the three different scores for each dimension. The correlations between the scores on the two dimensions, however, tend to be stronger for the network scores.

### 3.3. External Validation

The results of the zero-order correlations and of GGMs are reported in Figure 5 and Figure 6. Following a traditional approach to external validation, we would have inspected the bivariate, zero-order correlations (upper triangles) of WAS scores with each other variable in the matrix (columns 43 and 44 in Figure 5, columns 17 and 18 in Figure 6). This procedure would have led to conclude that in Cohort 1 the WAS scores tended to be positively associated with age, sleep issues, stress, effort, and overcommitment, and to be negatively associated with work capacity, job satisfaction, happiness, reward, and social support. Similarly, in Cohort 2, WAS scores tended to be positively associated with age, level of hostility towards elder workers, willingness to retire before 65, effort, overcommitment, anxiety, and depression and to be negatively correlated with perceived health condition, job satisfaction, happiness, working capacity, willingness to retire after 65, reward, and social support. 

Summarizing, the bivariate, zero-order correlations of the WAS scores with the other variables were generally weak or moderate (*r* ≤ |0.40|). Relevant for the construct validity of the scales, their correlations with other psychological measures, above all ERI-SF and DCSQ, were weak enough to suggest that the WAS scales pick up different working issues from these well-established measures. The pattern of correlations of the two WAS scales with the other variables was similar.

Such an approach, however, cannot rule out the possibility that some associations were spurious, i.e., dependent on some other variable in the correlation matrix. GGMs allow the researcher to control for spurious connections. In other words, coefficients in the lower triangle of matrices in Figure 5 and Figure 6 represent the partial correlation coefficients between the variables after conditioning on all other variables in the dataset, i.e., their direct associations. Rows 43 and 44 in Figure 5 and rows 17 and 18 in Figure 6 show these coefficients for the WAS scores. These correlations are inevitably very small in terms of effect size, but in this case, it should be considered whether they have been shrunk to zero by the algorithm or not. As a result, in Cohort 1 the dimension Working Conditions was positively associated with sex (i.e., men tended to endorse higher scores than women), age, sleep daytime dysfunction (PSQI-C7), daytime somnolence (BQ-C2 and ESS), happiness, effort, and stress, while it was negatively associated with dozing off while driving, working at night, reward, and overcommitment. The dimension Cognitive Demands was positively associated with age, being overweight, sleep latency, sleep daytime dysfunction, reward, and stress, while it was negatively correlated with the frequency with which participants tried to limit salt, sugar or fat in their meals; daytime somnolence; having been harassed at work; and working capacity. In Cohort 2, Working Conditions was positively associated with age, happiness, level of hostility against elderly workers, and negatively with sex (i.e., women scoring higher than men), perceived health status, reward, and overcommitment. Cognitive Demands was positively associated with age, level of hostility against elderly workers, and negatively with working capacity.

This approach also allows the detection of chains of indirect associations. For instance, the partial correlation matrix of data from Cohort 1 (Figure 5) shows a direct negative association of Working Conditions with ERI-SF-Reward (−0.16) and no association with job satisfaction. However, ERI-SF-Reward is directly and positively associated with job satisfaction (0.34), suggesting that Working Conditions is indirectly related to job satisfaction, and that any predictive effect from Working Conditions to job satisfaction is mediated by ERI-SF-Reward.

## 4. Discussion

The aim of this work was to show the application of a new and evolving methodological innovation, such as network psychometrics, to the development of a new measure of a psychological characteristic, work annoyance, that might play a key role in predicting psychological well-being (or lack thereof) in the workplace. 

Differently from classical psychometric models such as factor analysis and item response theory, which assume that an unobservable latent variable or construct, causes the observable behaviors and their pattern of covariation, network psychometrics considers psychological characteristics as states or stable organizations of dynamic components that can interact with and cause one another. This approach has important theoretical implications, since it addresses the long-standing issue of the actual existence of psychological constructs (see, e.g., [29]) and appears to be more consistent with the observation of the role that is played by observable behaviors. For instance, in psychopathology there appear to be many direct causal relationships between symptoms, and these sorts of associations can play an important generative role in the etiology of a disorder, other than accounting for the empirical covariance between symptoms (see, e.g., [98]). Recently, Christensen et al. [3] developed a set of statistical methods that allow researchers to investigate some basic psychometric properties of a psychological tool, such as redundancy, dimensionality, internal structure, and external validity, in this perspective. 

In this work, we showed an application of these methods to the development of a measure of work annoyance, i.e., the subjective evaluation of work characteristics as something causing vexation or nuisance in workers. Work annoyance pertains the workers’ actual perception of the characteristics of their job, without assuming that such characteristics are inherently positive or negative, as implied by some models such as the ERI or the DCS. This is consistent with research that found that some allegedly positive job characteristics, such as performing tasks that are cognitively challenging, are considered by some workers more of a nuisance than an opportunity [22] and that there is a positive association between skill discretion and job demands [23,24,25]. Grounding on common complaints that were reported by workers during compulsory annual medical examinations for occupational hazard, we identified nine job characteristics that could contribute to work annoyance, and we used them to develop the items of the Work Annoyance Scale (WAS). The scale was then administered to two cohorts of workers as part of their assessment, which provided the data used in this work. 

Initially, we performed “traditional” analyses, evaluating redundancies from the item zero-order correlation matrix, finding evidence for the optimal number of dimensions using the scree-test, parallel analysis, and MAP statistic, and testing the factor structure of the WAS using ESEM. We found evidence of a measurement model with two moderately correlated factors that replicated well across cohorts. The two factors were labelled as “Working Conditions” and “Cognitive Demands”, and omega coefficients indicated that they were adequately reliable. The zero-order correlations of their scores revealed a consistent pattern of association with background and health-related variables (i.e., weak-to-moderate positive associations with age, sleep disturbances) and workplace-related psychological dimensions (i.e., negative association with work capacity, job satisfaction, happiness, and reward; positive association with effort, overcommitment, stress, anxiety, and depression). However, one of the items (“work hard to solve a work problem”) had a substantial and nearly identical impact on both factors. In factor analysis, this result would raise the question of which scale the score of this item should contribute to, or might as well lead to the removal of the item from the scale.

We then used the function of the *EGAnet* package to test the psychometric properties of the WAS in a network psychometrics framework. The *UVA* function allowed us to test for redundancy. We found a redundancy chain between items 5, 6, and 3 in both cohorts, and between items 7 and 8 only in Cohort 2. Despite Christensen et al. [3] recommending removing all but one item from the questionnaire, keeping the item that represents the most general case of the attribute and/or has the most variance, or to combine items into a single variable, we suggest caution in doing so without a careful inspection of the content of redundant items, since statistical redundancy might not necessarily imply actual content redundancy. Sometimes redundant items are fundamentally the same item that are worded over and over in slightly different ways (e.g., ‘Hate being the center of attention’, ‘Make myself the center of attention’, ‘Like to attract attention’, and ‘Dislike being the center of attention’, as in [99]), but this might not always be the case. In this work, item 5 (learning how to use a new electronic device) and 6 (having to learn a foreign language) turned out to be statistically redundant, but they clearly tap into different issues.

We then tested the dimensional structure of the WAS using the *EGA* function. The results of this analysis were consistent with those of the dimensionality and ESEM analyses, as they indicated two dimensions (or communities) that grouped the items in the same way. However, this analysis allowed us to perform a direct test of which dimension item 9 should belong to. While it showed the same structural coefficients (i.e., the analogue of factor loadings, Table 1), the bootstrapped EGA analysis indicated that the item was consistently being identified within the cognitive demands dimension across the replicated samples. This result definitely indicated to retain the item and use it to assess only cognitive demands. The structural consistency coefficients approached 1.00, suggesting that each empirically-derived dimension was perfectly (i.e., exact same composition) recovered from almost all replicate bootstrap samples.

The overlap of the results of the two approaches is far from surprising, as the statistical equivalence between these models has already been demonstrated [100,101,102]. Consistent with this, we found near-perfect correlations (Table 3) between the sum scores, ESEM factor scores, and network scores that were obtained by multiplying the structural coefficients matrix by the observed data to derive a weighted composite for each dimension. However, the correlation between dimensions was substantially higher for network scores.

When investigating the external validity of a newly developed measure, one is usually concerned with the association of scores on the new measure with background variables and scores on other psychological measures that assess constructs in the same nomological network of the one under examination. Traditionally, this issue is addressed by computing and inspecting measures of bivariate association (zero-order correlations) and, although less often, specifying regression models for assessing the contribution of each potential predictor once controlling for all the others. When we used this approach, we found that WAS scale scores showed a pattern of association with health- and work-related background variables and psychological constructs that was consistent with previous studies: positive bivariate associations were found with issues and problems, such as sleep disturbance, stress, effort, and overcommitment, and negative associations were found with positive aspects of the job such as satisfaction, work capacity, reward, and social support. Without additional analyses, however, it was not possible to test whether these correlations could be spurious. We could have also specified regression models for evaluating the predicting power of these variables for the WAS scores, but as explained in the introduction, they would not have allowed us to map out collinearity and investigate predictive mediation. We addressed these issues using GGMs, and the coefficients that were reported in the lower triangles of Figure 5 and Figure 6 represent the residual association between two variables after controlling for all other information possible (i.e., conditional independence association): as a result, (potentially) spurious correlations were shrunk to zero through the regularization process performed by GGMs. The results did not substantially change but provided more convincing evidence of the discriminant validity of the WAS scores since they represented the strength of direct associations once controlling for all the other variables in the model. This allowed us to show the discriminant validity of the two scales of the WAS (Working Conditions and Cognitive Demands), as they were moderately related but also had a different pattern of association with the other variables, which was not so apparent when looking at zero-order correlations. Regarding the issue of whether a cognitively demanding working environment was associated with positive or negative outcomes, we could not find conclusive evidence, as the partial correlations of Cognitive Demands with the other psychological measures were mostly zero, except the negative correlation with work capacity, which was observed in both cohorts, and the negative correlation with ERI-SF-Reward, which was observed only in Cohort 1.

GGMs also offer the possibility to investigate indirect chains of associations, although care should be used in interpreting such results. Partial correlations can be indicative of potential causal pathways, but they do so when all the relevant variables are assumed to be observed, which is not the case here or in many psychological studies. Also, networks that are estimated on cross-sectional data like those that were used in this study are undirected networks. Thus, although non-zero partial correlation coefficients can be interpreted as a direct association between two variables that cannot be explained by any other variable in the network, the causal direction cannot be inferred—unless the assumptions of a Bayesian network analysis are met (see Introduction) and one can perform this analysis.

Other limitations should be considered when evaluating the results of this study. In order to meet the recommended sample sizes for network analyses, we used archival data that were not collected with research aims in mind, but, rather, had to comply with practical assessment objectives, as required by the Italian laws. This meant that the set of background and psychological variables that we could use for external validation could not be exhaustive of those characteristics that a research-oriented, hypothesis-based data collection would have included. For instance, a reviewer suggested to include a measure of burnout, which is a known consequence of job characteristics [103]. It is, therefore, desirable that future studies on the WAS would be designed to fill this gap. On a more general level, as suggested by a reviewer, when performing a GGM it should be considered that causal relationships may vary from one variable to another. So-called “control” variables that affect the relationship between two variables can be causes but also confounders, colliders, mediators, or proxies, and not all of them would require a direct statistical control [104]. As a result, researchers should explicitly define which variables are entered into the network and justify their relevance as potential contaminators on a theoretical ground prior to statistical computation—see also Briganti et al. [64] for details about causal inference.

As mentioned in the introduction, the WAS was developed before the COVID-19 pandemic. Therefore, it might be argued that its content validity might need to be improved as organizations had to make changes in terms of work setting, work environment, and procedures and the ongoing transition to new norms has not been without consequences on workers’ well-being [105]. For instance, the pandemic has made working from home more and more common and its effects on workers’ well-being can be cause of concern (see, e.g., [106,107,108]). A future revision of the WAS might thus include a question about the annoyance that is associated with working from home. As a further limitation, the data were collected in an Italian context: as such, participants shared societal values of the Italian culture, and this is known to affect not only social expectations and behavioral norms [109], but also the association of work strain with workers’ attitudes and intentions and the way in which workers interpret their work experiences [110]. Therefore, we suggest to investigate the cross-cultural generalizability of these results by means of national adaptation studies of the WAS.

## 5. Conclusions

In this work, we showed an application of a recent advanced analytical methodology, network psychometrics, in the evaluation of the psychometric properties of a newly developed psychological measure. Specifically, we highlighted how this approach can be useful in addressing substantively relevant questions that might not be appropriately addressed by traditional methods (e.g., which dimension an item with cross-loadings actually belongs to? Which background and psychological variables the scale scores are non-spuriously associated to?) while still allowing the researcher to assess dimensionality and structural consistency of the scales. However, as noted by a reviewer, it would be too simplistic to think that EGA relieves the researcher of any decision in all situations. Sometimes items may have genuine factor loadings on several factors because they operationalize several factors at once. Even if EGA allocates this item to one particular factor, it does not necessarily mean that this item does not depend on the other factor(s). As a result, the strict allocation of items to concrete factors should not be considered a panacea. In most cases, one might want to consult both factor analysis and network analysis to make a deliberate judgment.

Using the network approach, we could also provide evidence that the Work Annoyance Scale (WAS) is a psychometrically sound tool for the assessment of work annoyance, i.e., the subjective evaluation of work characteristics as something causing vexation or nuisance. Due to its brevity and ease of administration, it can be a useful add-on in surveys, assessments, and/or research in organizational contexts.

## Figures and Tables

**Figure 1 ijerph-19-09376-f001:**
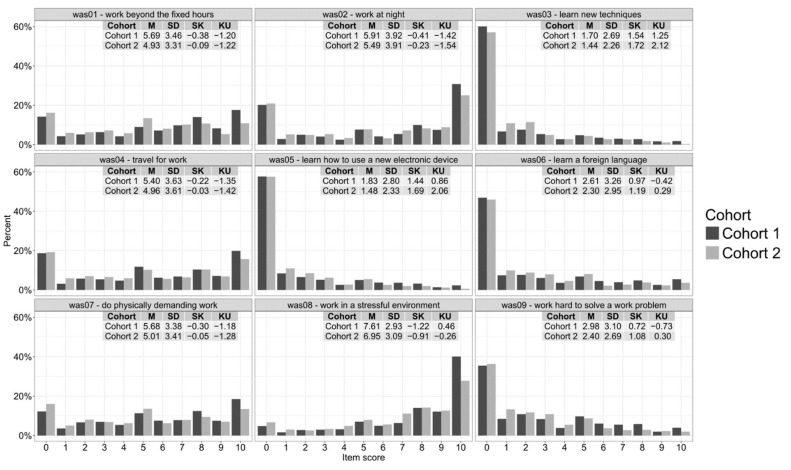
Score distributions and descriptive statistics for the items of the Work Annoyance Scale in the two cohorts (Cohort 1 *n* = 2226; Cohort 2 *n* = 655). *M* = mean; *SD* = standard deviation; *SK* = skewness; *KU* = kurtosis.

**Figure 2 ijerph-19-09376-f002:**
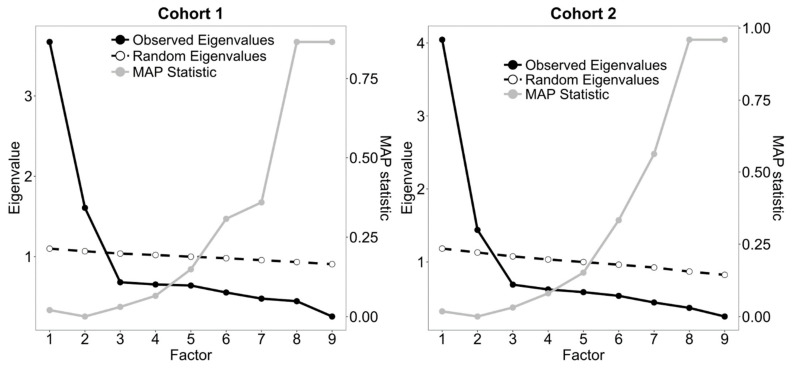
Scree-, parallel analysis, and minimum average partial (MAP) correlation statistic plots for the Work Annoyance Scale in the two cohorts (Cohort 1 *n* = 2226; Cohort 2 *n* = 655).

**Figure 3 ijerph-19-09376-f003:**
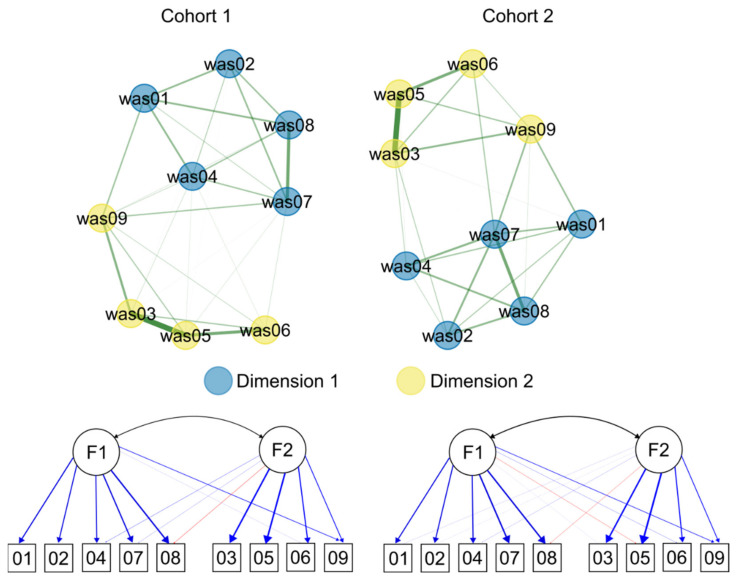
Dimensions that were identified by Exploratory Graph Analysis (EGA, **top**) and Exploratory Structural Equation Modeling (ESEM, **bottom**). In the EGA plot, the color of the nodes represents the dimension and the thickness of the lines represents the magnitude of the partial correlations. In the ESEM plot, the thickness of the lines represents the magnitude of the pattern coefficients (blue = positive, red = negative) as in Table 2.

**Figure 4 ijerph-19-09376-f004:**
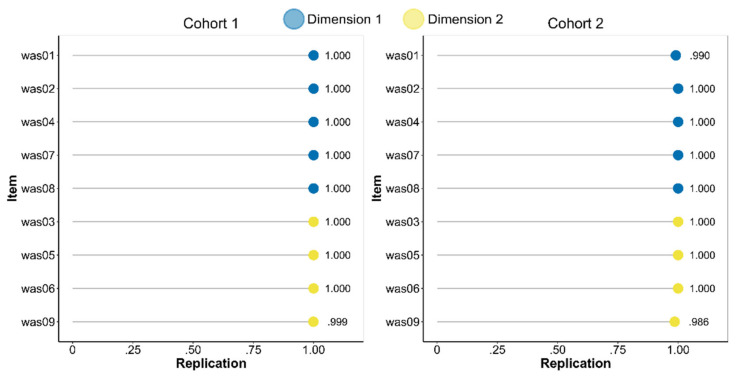
Items’ proportions of replication in the original dimension as specified by the Exploratory Graph Analysis.

**Figure 5 ijerph-19-09376-f005:**
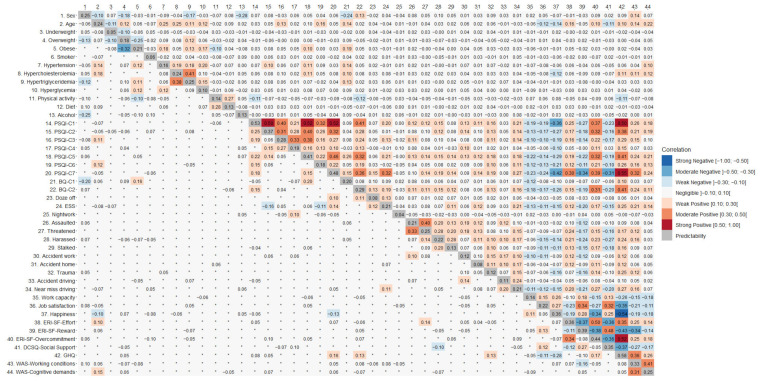
Correlations (upper triangle), partial correlations (lower triangle), and predictability values (diagonal) for the external validation of the Work Annoyance Scale (WAS) in Cohort 1 (*n* = 2018). PSQI: Pittsburgh Sleep Quality Index; BQ: Berlin Questionnaire; ESS: Epworth Sleepiness Scale; ERI-SF: Effort-Reward Imbalance Questionnaire-Short Form; DCSQ: Demand-Control-Support Questionnaire; GHQ: General Health Questionnaire. The symbol “°” indicates a partial correlation shrunk to zero. In the legend, the outward facing brackets indicate that the value is not included in the interval, while the inward facing brackets indicate that the value is not included in the interval.

**Figure 6 ijerph-19-09376-f006:**
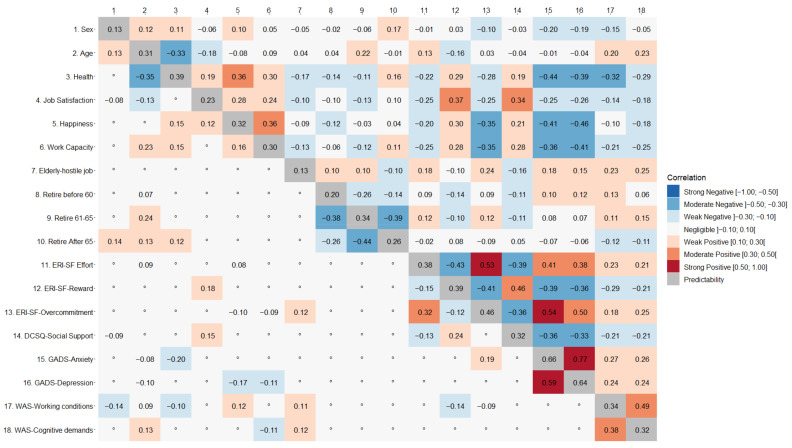
Correlations (upper triangle), partial correlations (lower triangle), and predictability values (diagonal) for the external validation of the Work Annoyance Scale (WAS) in Cohort 2 (*n* = 655). ERI-SF: Effort-Reward Imbalance Questionnaire-Short Form; DCSQ: Demand-Control-Support Questionnaire; GADS: Goldberg Anxiety and Depression Scale. The symbol “°” indicates a partial correlation shrunk to zero. In the legend, the outward facing brackets indicate that the value is not included in the interval, while the inward facing brackets indicate that the value is not included in the interval.

**Table 1 ijerph-19-09376-t001:** Descriptive statistics of the variables used in this work (1/2) (2/2).

Variable	Cohort 1 (*n* = 2226)	Cohort 2 (*n* = 655)
Age (M ± SD, range)	47.61 ± 9.40 (19–75)	44.00 ± 12.19 (20–67)
Female (%)	32.34%	63.49%
Working at night (%)	8.89%	NA
ERI-SF—Effort (M ± SD, range, ω, AVE)	2.25 ± 0.77 (1–4), 0.85 [0.84, 0.86], 0.65	2.34 ± 0.75 (1–4), 0.83 [0.81, 0.86], 0.63
ERI-SF—Reward (M ± SD, range, ω, AVE)	2.50 ± 0.55 (1–4), 0.77 [0.76, 0.78], 0.37	2.39 ± 0.54 (1–4), 0.76 [0.73, 0.79], 0.36
ERI-SF—Overcommitment (M ± SD, range, ω, AVE)	2.69 ± 0.62 (1–4), 0.83 [0.82, 0.84], 0.46	2.71 ± 0.63 (1–4), 0.84 [0.83, 0.86], 0.49
DCSQ—Social support (M ± SD, range, ω, AVE)	3.20 ± 0.56 (1–4), 0.89 [0.88, 0.90], 0.59	3.30 ± 0.53 (1–4), 0.89 [0.88, 0.91], 0.60
GHQ (M ± SD, range, ω, AVE)	2.00 ± 0.48 (1–4), 0.93 [0.92, 0.93], 0.54	NA
Working capacity (M ± SD, range)	8.43 ± 1.87 (0–10)	8.03 ± 1.67 (0–10)
WAS—Working conditions (M ± SD, range, ω, AVE)	6.06 ± 2.51 (0–10), 0.77 [0.75, 0.78], 40	5.47 ± 2.60 (0–10), 0.80 [0.78, 0.83], 0.45
WAS—Cognitive demands (M ± SD, range, ω, AVE)	2.28 ± 2.35 (0–10), 0.81 [0.79, 0.82], 51	1.90 ± 2.05 (0–10), 0.81 [0.78, 0.83], 0.51
Job satisfaction (M ± SD, range)	4.52 ± 1.49 (1–7)	4.40 ± 1.55 (1–7)
Happiness (M ± SD, range)	6.97 ± 1.86 (0–10)	7.09 ± 1.91 (0–10)
GADS—Anxiety (M ± SD, range, ω, AVE)	NA	3.84 ± 2.96 (0–9), 0.92 [0.91, 0.93], 0.61 0.61.6161
GADS—Depression (M ± SD, range, ω, AVE)	NA	2.69 ± 2.45 (0–9), 0.92 [0.91, 0.93], 0.58
Physically assaulted at work (%)	9.30%	NA
Threatened at work (%)	14.87%	NA
Harassed at work (%)	18.69%	NA
Stalked at work (%)	5.88%	NA
Accident at work (%)	7.50%	NA
Accident at home (%)	11.86%	NA
Trauma (e.g., death of a beloved one) (%)	32.08%	NA
Accident while driving (%)	7.95%	NA
Came close to having an accident while driving (%)	25.43%	NA
Sleeping while driving		
Never or almost never	93.22%	NA
1–2 times a month	3.86%	NA
1–2 times a week	1.48%	NA
3–4 times a week	0.63%	NA
Almost every day	0.81%	NA
PSQI—Component 1: Subjective sleep quality (M ± SD, range)	1.07 ± 0.81 (0–3)	NA
PSQI—Component 2: Sleep latency (M ± SD, range)	0.64 ± 0.68 (0–3)	NA
PSQI—Component 3: Sleep duration (M ± SD, range)	1.45 ± 1.00 (0–3)	NA
PSQI—Component 4: Habitual sleep efficiency (M ± SD, range)	0.27 ± 0.68 (0–3)	NA
PSQI—Component 5: Sleep disturbances (M ± SD, range)	1.17 ± 0.64 (0–3)	NA
PSQI—Component 6: Use of sleeping medication (M ± SD, range)	0.25 ± 0.74 (0–3)	NA
PSQI—Component 7: Daytime dysfunction (M ± SD, range)	0.79 ± 0.77 (0–3)	NA
BQ—Category 1: Snoring (M ± SD, range)	23.36%	NA
BQ—Category 2: Daytime somnolence (M ± SD, range)	18.69%	NA
ESS (M ± SD, range, omega, AVE)	0.73 ± 0.52 (0–3), 0.88 [0.87, 0.88], 0.49	NA
Underweight (%)	2.64%	NA
Normal weight (%)	54.69%	NA
Overweight (%)	29.63%	NA
Obesity (%)	13.05%	NA
Hypertension (%)	16.58%	NA
Hypercholesterolemia (%)	27.58%	NA
Hypertriglyceridemia (%)	10.29%	NA
Hyperglycemia (%)	5.03%	NA
Weekly physical activity		
Never	48.61%	NA
Once	16.85%	NA
Twice	16.85%	NA
Three or more times	17.70%	NA
Low-fat, low-sugar, low-salt diet		
Never	19.27%	NA
In some meals	28.98%	NA
In most meals	33.38%	NA
Every meal	18.37%	NA
Daily alcohol consumption (units)		
None	59.30%	NA
1–7	36.97%	NA
8–16	2.96%	NA
17+	0.76%	NA
Smoking (%)	33.85%	NA
Desired retirement age: Before 60 (%)	NA	8.84%
Desired retirement age: Between 60 and 65 (%)	NA	41.80%
Desired retirement age: After 65 (%)	NA	17.52%
Desired retirement age: Don’t know (%)	NA	31.83%
Elder-worker-hostile environment		
Not at all	NA	7.12%
A little	NA	22.33%
Somewhat	NA	41.75%
Much	NA	20.23%
Very much	NA	8.58%
Perceived health condition		
Very bad	NA	3.76%
Bad	NA	31.77%
Neither good nor bad	NA	7.36%
Good	NA	43.97%
Very good	NA	13.15%

Note: M = mean; SD = standard deviation; ω: omega reliability coefficient, computed as omega total with the scale structure function from the ufs package in Ref. [66]; AVE = average variance extracted; NA: data not available; ERI-SF: Effort-Reward Imbalance Questionnaire-Short Form; DCSQ: Demand-Control-Support Questionnaire; GHQ: General Health Questionnaire; WAS: Working Annoyance Scale; GADS: Goldberg Anxiety and Depression Scale; PSQI: Pittsburgh Sleep Quality Index; BQ: Berlin Questionnaire; ESS: Epwoth Sleepiness Scale.

**Table 2 ijerph-19-09376-t002:** Factor loadings (pattern coefficients) from the Exploratory Structural Equation Modeling (ESEM) analyses and structural coefficients (network loadings) from the network analysis.

	Pattern Coefficients		Network Loadings	
Item	F1	F2	D1	D2
was01	0.62 0.59	0.00 0.04	0.31 0.26	0.08 0.10
was02	0.57 0.59	0.00 0.02	0.29 0.27	0.01 0.05
was03	0.00 0.02	0.84 0.82	0.04 0.09	0.48 0.46
was04	0.54 0.60	0.12 0.06	0.27 0.29	0.09 0.05
was05	−0.01 −0.11	0.87 0.94	0.04 0.01	0.51 0.50
was06	0.04 0.09	0.60 0.58	0.04 0.06	0.26 0.26
was07	0.67 0.80	0.07 0.00	0.35 0.42	0.12 0.15
was08	0.78 0.79	−0.15 −0.10	0.39 0.40	0.02 0.02
was09	0.32 0.30	0.44 0.43	0.19 0.18	0.22 0.22
F1 with F2	0.41 0.53			

Note: in each column, values for Cohort 1 (*n* = 2226) are on the left, values for Cohort 2 (*n* = 655) are on the right. F1/2 = Factors: D1/2: Dimensions. F1 with F2: correlation between factors.

**Table 3 ijerph-19-09376-t003:** Correlations among the observed scores, factor scores from the exploratory structural equation modeling (ESEM), and network scores in Cohort 1 (*n* = 2226, lower triangle) and Cohort 2 (*n* = 655, upper triangle) for Dimension 1 (D1, “Working conditions”) and Dimension 2 (D2, “Cognitive demands”) of the Work Annoyance Scale.

	1	2	3	4	5	6
1. Observed D1		0.976	0.987	0.489	0.506	0.665
2. ESEM D1	0.976		0.997	0.598	0.590	0.748
3. Network D1	0.986	0.994		0.593	0.591	0.746
4. Observed D2	0.417	0.516	0.536		0.957	0.966
5. ESEM D2	0.399	0.478	0.500	0.962		0.968
6. Network D2	0.558	0.636	0.654	0.974	0.976	

## Data Availability

The data that are presented in this study are available on request from the corresponding authors. The data are not publicly available as participants did not provide explicit consent for their data to be shared.

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
