# Peer review of "Development and Validation of a New Measure of Work Annoyance Using a Psychometric Network Approach"

_ijerph, 2022, doi:10.3390/ijerph19159376_

Round 1
Reviewer 1 Report
Brief summary
The present study evaluated the psychometric properties of a newly developed Work Annoyance Scale (WAS)—implementing both latent variable framework and network psychometric approach—and demonstrated its soundness as a valuable research tool in future research in organizational contexts.
Broad comments
The manuscript is well-prepared and has many positives. The manuscript’s literature review is thorough and up to the point. The research rationale is clearly communicated and properly substantiated. The sample size is large enough for having adequate statistical power for testing all hypotheses. Moreover, the authors explicitly defined the minimal sample size in accordance with methodological recommendations. In addition, the authors had a cross-validation sample which allowed them to replicate their findings across different samples. The data analytical strategy is outlined perfectly well and supported by access to the R code. Nevertheless, I have some minor comments on various parts of the manuscript that are outlined in the next section. I sincerely hope that at least some of them will be of value during the revision. All references below are provided primarily to unwrap and substantiate my point of view and should not be considered as suggestions for their addition to the manuscript unless stated otherwise. PDF document is also attached in case if hyperlinks for cited references won't work.
Specific comments
Title
Q1. Lines 2–3. Please remove needless hyphens from the word “Annoyance.”
Introduction
Q2. Although section 1.2 is surely well-written, I guess it can be splinted into several subsections to make it easier to follow the authors’ narration. For instance, one can divide section 1.2 into two: (1.2.1) latent variable approach and (1.2.2) network approach. Next, one can divide subsection 1.2.2 into four: (a) redundancy, (b) dimensionality, (c) internal consistency, and (d) external validation. However, I don’t consider it a crucial issue and prefer leaving it to the authors’ opinion.
Q3. Lines 99–100. I have two questions related to the fact that the authors developed both Italian and English versions of the Working Annoyance Scale (WAS). First, am I correct that psychometric validation was conducted only for the Italian version? Second, what was the rationale for developing scale in both Italian and English? I may be wrong, but perhaps the authors would like to facilitate the scale dissemination by providing its English version. If so, it would be beneficial to report how the authors determined language equivalence between the Italian and English versions (e.g., backward translation or something different). Information about scale translation is tangential and—if the authors will find it appealing—they can report additional details not in the main text but in the supplementary materials. In sum, I suggest reporting explicitly which version—Italian or English (maybe both?)—was validated and how the translation process was administered. In addition, if the English version was not validated in the English-speaking population, one should put a disclaimer that the English version still awaits its empirical validation.
Q4. Lines 218–220. “EGA initially estimates a Gaussian graphical model (GGM) [45,46], using the graphical least absolute shrinkage and selection operator (GLASSO [47])…” Strictly speaking, EGA with GLASSO is one but not the only option. In some instances, one can perform EGA using triangulated maximally filtered graph approach (TMFG; see Golino et al., 2020).
Materials and Methods
Q5. In Table 1, internal consistency reliability is reported as omega coefficients. Please could the authors specify which omega coefficient was computed? The problem is that several omega coefficients are available (e.g., hierarchical, total, etc.; Flora, 2020; Kelly & Pornprasertmanit, 2016).
Q6. Lines 434–443. Is it possible to separate the description of measures applied in Cohort 1 from those in Cohort 2? For example, one can put them in different subsections. The idea is to make it visually clear where each Cohort’s measures are described. At the same time, I don’t see it as a crucial issue and prefer leaving it to the authors’ opinion.
Q7. Line 455. “We initially performed the traditional analyses of dimensionality of the item pool…” It would be better to identify that it’s about the item pool for the Work Annoyance Scale. Otherwise, a reader may be a bit confused.
Q8. Lines 460–463. Two issues. First, parallel analysis was performed on the matrix of Pearson correlations or polychoric correlations? Second, parallel analysis was performed with Principal Component Analysis (PCA) or factor analytical approach? If factor analysis was used, please report the estimation algorithm and the choice rationale (i.e., why this particular estimation method was chosen?).
Q9. Lines 468–469. “Once determined the optimal number of factors, we used ESEM to investigate the factor structure.” It would be better to identify that it’s about factor structure for the Work Annoyance Scale. Otherwise, a reader may be a bit confused.
Q10. Line 478. “90% confidence interval” should be “90% confidence intervals.”
Q11. Lines 483–484. “Finally, we tested the external validity of the newly developed scales using network analysis.” Have the authors developed several new scales? In fact, I thought there was only one new scale (i.e., Work Annoyance Scale), was not it?
Results
Q12. Lines 522–524. Is it possible to accompany goodness-of-fit indices with values of Standardized Root Mean Square Residual (SRMR)? I suppose it’s a valuable indicator that provides useful information on global fit without overlapping with other indices. But I don’t insist on it and prefer leaving it to the authors’ judgment.
Q13. Line 524. “The factor loadings and correlation...” Maybe “interfactor correlation” or “correlation between factors” would be more accurate than simply “correlation”?
Q14. I wonder if it is possible to replace “ESEM loadings” with “ESEM pattern coefficients” and “Structural coefficients” with “Network loadings” in Table 2. The term “loading” is ambiguous and can refer to both pattern and structural coefficients in the factor analytical framework (Kline, 2015), whereas “network loadings” is more consistent with terms used in Christensen and Golino (2021). However, I leave it to the authors’ judgment.
Q15. Lines 563–564. Was EGA performed on the matrix of Pearson correlations?
Q16. Line 584. “Omega reliability coefficients were…” Please specify which omega coefficient was computed (similar to Q5).
Q17. Line 603. “…almost perfect overlap among the three difference scores…” Am I right that the authors meant “different scores” instead of “difference scores”? If so, please correct it accordingly.
Q18. Line 611. What are GMMs? Should it be GGM instead?
Q19. Lines 612–614. “Following a traditional approach to external validation, we would have inspected the 612 bivariate, zero-order correlations (upper triangles) of WAS scores with each other variable 613 in the matrix (columns 43 and 44 in Figure 5, columns 17 and 18 in Figure 6).” If I took it right, the authors haven’t got predetermined hypotheses on correlations between WAS subscales and other variables. Thus, they investigated how WAS subscales are related to any of the variables. If so, I wonder if the authors considered introducing some kind of correction on multiple hypotheses testing. I would like to make it clear that I don’t argue for introducing one. Moreover, I would expect that due to the large sample size and nominal alpha level (α = .05) most notable correlations will survive even after applying corrections. Nevertheless, I would be grateful to the authors if they could present their position on the issue, and what was their rationale to introduce/to dismiss corrections on multiple hypotheses testing.
Q20. Line 624. I guess “weak o moderate” should be “weak or moderate.”
Q21. Lines 629–655. I don’t argue against the rationale to control for spurious correlations. However, I wonder about the extent to which the approach is viable given that causal relationships may vary from one variable to another. For example, third variables—contaminating the relationship between two variables—can be confounders, colliders, mediators, or proxies, and not all of them would require a direct statistical control (see Wysocki, Lawson, & Rhemtulla, 2022). Moreover, one has to explicitly define which variables are entered into the network and justify their relevance as potential contaminators on the theoretical ground prior to statistical computation. Otherwise, a ubiquitous statistical control will result in a substantial coefficients bias, won’t it? If the authors find my concerns relevant, it would be important to also make some adjustments to the discussion of these findings written on lines 761–771. If not, feel free to overrule me.
Q22. Legends in Figure 5 and 6 has errors in depicting correlation segments (i.e., wrongly posed brackets). For example, “]-.50; -.30]” should be “[-.50; -.30].”
Discussion
Q23. Line 770. “…more convincing evidence of the validity of WAS…” The validity term is too broad. Maybe discriminant validity would be more relevant here?
Conclusions
Q24. Lines 813–814. “(e.g., which dimension an item with cross-loadings actually belongs to?” I don’t argue against the fact that in some instances, EGA is superior to more traditional factor analytical approaches to the allocation of items to factors. However, I think it would be too simplistic to expect that EGA doesn’t require any judgments in all situations. Why? In some cases, items can have genuine factor loadings on several factors because they do characterize several factors at once. Even if EGA allocates this item to one particular factor, it doesn’t follow that this item won’t depend on the other factor. Thus, strict allocation of items to concrete factors is not a panacea. In most cases, one will need to consult both factor analysis and network analysis to make a deliberate judgment.
Reviewer 2 Report
A generally good paper - For consideration, I recommend the inclusion of research which briefly presents / discusses the conflicting and divergent perspectives in the literature (2-3 paragraphs), updated the Abstract to briefly note limitations, and then include the limitations following your Methodology section as opposed to the back end of the paper. Finally, I would recommend that you have a full professional edit to ensure scholarly flow and readability is evident throughout your paper.
If you take the time and effort to incorporate the spirit and intent of the suggested revisions to your paper, you will take your paper to the next step – that is publication and a following for your research.
Overall an interesting paper, which presents, and discusses the Work Annoyance Scale which was used to determine the level of annoyance that Italian workers feel about certain aspects of the job through an evaluation of its psychometric properties.
From this perspective, the paper will garner interest from academia. From an academic perspective the mix of current (less than 5 year peer reviewed research) presented and discussed in the paper is overall representative of the current literature. However, as an opportunity to demonstrate inclusiveness, there is need for a brief (2-3 paragraphs) on divergent and conflicting perspectives within the literature so as to avoid the narrow focus perception which many academic would consider as selective literature to support research agenda. So to avoid this shortcoming and to garner academic community interest, it is necessary to extend the research to add the conflicting and divergent perspectives from the literature to ensure the paper’s overall comprehensiveness and currency to the intended audience.
From a methodology perspective, the paper is generally well designed and appropriate – the exception is the lack of limitations noted in the Abstract; and this may very well reflect my own personal preferences. However, in defense of this posture, the identification of limitations at the front (Abstract and at the end of the Methodology Section) enables the reader audience to understand the limitations and any caveats up front, and prior to reading the paper through and then findings limitations at the back end of the paper. It is imperative that reader(s) have the opportunity to place the paper into a relevant context in relation to what is stated / proposed by the author(s).
The results are generally presented in a clear and concise manner; there is evidence of analysis evident; the inclusion of Tables and Figures are an effective visual. As well, the author(s) have developed an acceptable linkage between the results and conclusions noted. The points noted in paper are tied together into a final coherent picture. It is evident that the author(s) have an excellent understanding of the subject area.
This is a solid paper in many respects, since it provides several opportunities for continued research in the subject area with the possibility of different streams within the research area, while providing further avenues of research potential. With respect to the practical application of the research, it presents an opportunity to enhance the depth, breadth and understanding of factors related to work annoyance.
The writing quality of the paper requires a professional edit to ensure scholarly flow and readability.
Comments to the Author
A generally good paper - For consideration, I recommend the inclusion of research which briefly presents / discusses the conflicting and divergent perspectives in the literature (2-3 paragraphs), updated the Abstract to briefly note limitations, and then include the limitations following your Methodology section as opposed to the back end of the paper. Finally, I would recommend that you have a full professional edit to ensure scholarly flow and readability is evident throughout your paper.
If you take the time and effort to incorporate the spirit and intent of the suggested revisions to your paper, you will take your paper to the next step – that is publication and a following for your research.
Reviewer 3 Report
Dear authors,
thank you for allowing me to review your work, which I found quite organised and well done. The contribution concerns the development of a measure, the Work Annoyance Scale (WAS), to assess the level of annoyance that workers express about certain aspects of their work, evaluating its psychometric characteristics. The results show a two-dimensional structure, validated on two datasets.
1. The introductory part, with the models mentioned, is fine. I recommend, for the sake of completeness, also mentioning the Job Demands-Job Resources model, as annoyance could also be considered as a job demand.
2. Honestly, I find the methodology part rather long and redundant, in the sense that it adds little to what has been implemented. What I mean is that it is a well-done part, but for the purposes of the magazine it is a bit too much. It would be a well-drawn-out part for a psychometric journal, but on this type of journal it is also fine to reduce it to the essential prodromal parts for the analyses you have performed.
3. Can you add the skewness and kurtosis indices?
4. Reliability and validity indices for all variables are missing. Insert at least Cronbach's alpha (or McDonald omega) and AVE for constructs.
5. Psychometric analyses are very good. However, a representation of the final scale with dimensions and manifest indicators according to latent variables is missing.
Reviewer 4 Report
First of all, congratulations for the work carried out, it is a very interesting research and the methodology of the study is very well defined and developed. After reviewing the manuscript, it could be improved in the following points:
The introduction is too long, and the reader is lost in understanding the justification of the study. Due to the subject matter, the introduction should show the concept of "burnout" as an occupational problem and questionnaires such as Maslach's MBI, which are frequently used in research.
Due to the period of the study, the introduction should address the changes in the workplace as a consequence of the pandemic, which are discussed in the limitations of the study.
Best regards.
